# Efficient and simultaneous capture of iodine and methyl iodide achieved by a covalent organic framework

Yaqiang Xie[1], Tingting Pan[1], Qiong Lei [1], Cailing Chen[1], Xinglong Dong [1], Youyou Yuan[2], Walid Al Maksoud[3], Long Zhao[4], Luigi Cavallo [3], Ingo Pinnau [1] & Yu Han [1,3 ✉]

Radioactive molecular iodine ($I_2$) and organic iodides, mainly methyl iodide ($CH_3I$), coexist in the off-gas stream of nuclear power plants at low concentrations, whereas few adsorbents can effectively adsorb low-concentration $I_2$ and $CH_3I$ simultaneously. Here we demonstrate that the $I_2$ adsorption can occur on various adsorptive sites and be promoted through intermolecular interactions. The $CH_3I$ adsorption capacity is positively correlated with the content of strong binding sites but is unrelated to the textural properties of the adsorbent. These insights allow us to design a covalent organic framework to simultaneously capture $I_2$ and $CH_3I$ at low concentrations. The developed material, COF-TAPT, combines high crystallinity, a large surface area, and abundant nucleophilic groups and exhibits a record-high static $CH_3I$ adsorption capacity ($1.53\,g \cdot g^{-1}$ at 25 °C). In the dynamic mixed-gas adsorption with 150 ppm of $I_2$ and 50 ppm of $CH_3I$, COF-TAPT presents an excellent total iodine capture capacity ($1.51\,g \cdot g^{-1}$), surpassing various benchmark adsorbents. This work deepens the understanding of $I_2/CH_3I$ adsorption mechanisms, providing guidance for the development of novel adsorbents for related applications.

[1] Advanced Membranes and Porous Materials (AMPM) Center, Physical Science and Engineering Division, King Abdullah University of Science and Technology (KAUST), Thuwal, Saudi Arabia. [2] Imaging and Characterization Core Lab, King Abdullah University of Science and Technology (KAUST), Thuwal, Saudi Arabia. [3] KAUST Catalysis Center, Physical Science and Engineering Division, King Abdullah University of Science and Technology (KAUST), Thuwal, Saudi Arabia. [4] State Key Laboratory of Advanced Electromagnetic Engineering and Technology, School of Electrical and Electronic Engineering, Huazhong University of Science and Technology, Wuhan 430074, China. ✉email: yu.han@kaust.edu.sa

Nuclear reactors have been continuously providing ~10% of the world's energy over the last decade[1]. As a sustainable and low-carbon energy supply, nuclear energy is expected to play a more important role in the future[2–4]. However, safety concerns still challenge its operation. One of the major safety issues is the volatile radioactive waste produced during the reprocessing of spent nuclear fuels, which primarily consists of radionuclides, such as $^{129}$I and $^{131}$I in the form of molecular iodine ($I_2$) or organic iodides (e.g., methyl iodide ($CH_3I$) and ethyl iodide)[4–8]. These compounds are harmful to the environment ($^{129}$I has an extremely long half-life of approximately $1.57 \times 10^7$ years) or severely affect human metabolism by damaging the thyroid gland, and must be removed before the off-gas is discharged[9–11].

Compared with traditional liquid scrubbing processes to capture radioactive iodine, adsorption-based processes require a simpler operation and lower maintenance costs and avoid highly corrosive solutions[6]. Therefore, researchers have increasingly focused on the development of various adsorbents for iodine capture, including materials containing silver (Ag)[12–14], ceramics[13,15,16], zeolites[17,18], aerogels[19–21], metal-organic frameworks[8,22–26], and conjugated polymers[27–31]. Most of these studies have focused on the adsorption capacity of the developed adsorbent for $I_2$, whereas only a few studies have addressed the capture of $CH_3I$, and even fewer studies have examined the simultaneous capture of $I_2$ and $CH_3I$. Given that radioactive molecular iodine and organic iodides coexist in off-gas streams, it is particularly important to develop adsorbents that can capture them simultaneously and efficiently.

Various strategies have been adopted to promote the adsorption of iodine species. For $I_2$, effective strategies include the following: (i) using adsorbents containing Ag to precipitate $I_2$ in the form of silver iodide (AgI)[2,12,13], (ii) introducing electron-rich heteroatoms (e.g., nitrogen (N), sulfur (S), and oxygen(O)) or π-donors (e.g., double/triple bonds, benzene rings, and other aromatic compounds) in adsorbents to adsorb electron-deficient $I_2$ by forming charge-transfer complexes[32–41], and (iii) modifying the adsorbent with ionic groups (e.g., $[RN-(CH_3)_3]^+\cdot Br^-$) to adsorb $I_2$ via Coulomb interactions[42]. Compared with $I_2$, $CH_3I$ is more difficult to capture because of the weaker intermolecular forces[43]. Currently, the capture of $CH_3I$ is primarily achieved either through catalytic decomposition on adsorbents containing Ag to form AgI[13,44–46] or through an N-methylation reaction on nucleophilic N sites to form pyridinium[34] or quaternary ammonium salts[1,31,47–49]. Based on these insights, we speculate that adsorbents containing abundant Ag sites or nucleophilic N sites may exhibit high capture capacity for both $I_2$ and $CH_3I$. Given the high cost and poor recyclability of Ag-based adsorbents, developing N-rich adsorbents is a better choice for simultaneously capturing $I_2$ and $CH_3I$.

In addition to the characteristics of adsorption sites that determine binding strength, the density of the adsorption sites and the textural properties (e.g., surface area, pore size, and pore volume) of the adsorbent are crucial because these factors collectively determine the number of accessible adsorption sites (i.e., the adsorption capacity) and adsorption kinetics. Therefore, an ideal adsorbent for simultaneously capturing $I_2$ and $CH_3I$ should possess a high concentration of nucleophilic N sites along with a large surface area and a highly open porous structure.

As an emerging class of porous materials, covalent organic frameworks (COFs) provide an ideal platform for developing high-performance adsorbents because their porous structures and surface functionalities can be easily engineered to meet the requirements of specific applications[50–52]. Several COFs have been prepared as adsorbents for $I_2$ capture, in which the binding sites are π-conjugated moieties, various N-containing functional groups, and ionic groups[34,35,38,42]. Although some of these COFs exhibited high $I_2$ adsorption capacities in the measurement performed in a static closed system with high partial pressure of $I_2$, their performance for low-concentration $I_2$ capture under a dynamic condition was not measured. More importantly, like other recently developed adsorbents, the $CH_3I$ adsorption properties of these COFs have not been investigated.

To the best of our knowledge, there is only one COF material (SCU-COF-2) evaluated for both $I_2$ and $CH_3I$ adsorption[34]. In SCU-COF-2, there are pyridine and imine moieties, which bind to $I_2$ and $CH_3I$ through charge-transfer interaction and N-methylation reactions, respectively. At room temperature, SCU-COF-2 adsorbed 0.979 g g$^{-1}$ $I_2$ from flowing $N_2$ (carrier gas) containing 400 ppm of $I_2$ or 0.564 g g$^{-1}$ of $CH_3I$ from flowing $N_2$ containing 200,000 ppm of $CH_3I$. These moderately high adsorption capacities were obtained from single-component measurements, whereas the adsorption performance of SCU-COF-2 in the coexistence of $I_2$ and $CH_3I$ has not been explored.

In this report, we designed and synthesized two COFs, namely COF-TAPB and COF-TAPT, for the simultaneous capture of $I_2$ and $CH_3I$. The framework of COF-TAPB was constructed through imine linkages formed between two monomers, tris(4-formylphenyl)amine (TFPA) and 1,3,5-tris(4-aminophenyl)benzene (TAPB); COF-TAPT is a structural analog of COF-TAPB, constructed from TFPA and 2,4,6-tri(4-aminophenyl)-1,3,5-triazine (TAPT) (see Fig. 1). Through a literature search, we found

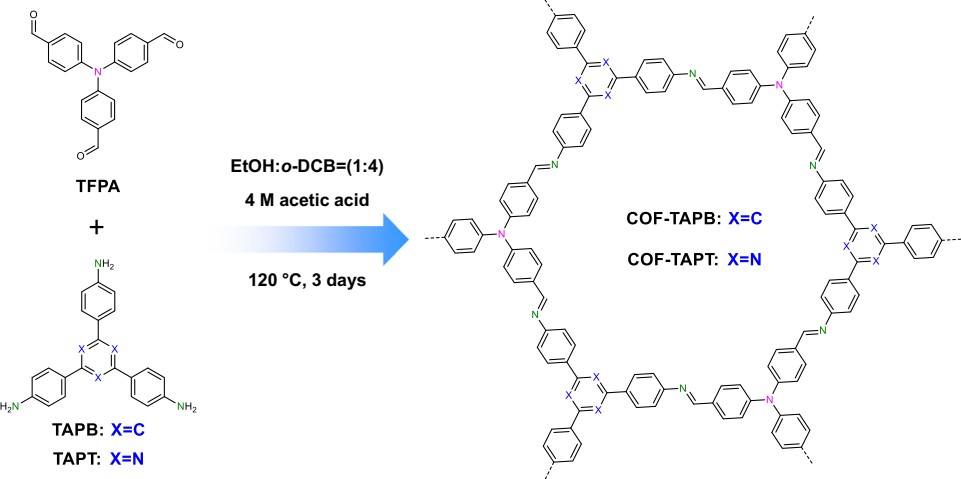

**Fig. 1 Schematic illustration of the synthesis of COF-TAPT and COF-TAPB.** In the structural model, different N sites are marked with different colors.

that these two COF materials have previously been synthesized for $CO_2$ adsorption and photocatalytic hydrogen evolution[53–55], whereas the synthetic methods are not exactly the same as those used in this study. The two COFs exhibit the same crystal structure and textural properties, being different only in N content, allowing the investigation of the role of N in the adsorption of $I_2$ and $CH_3I$.

We evaluated the adsorption properties of the two COFs for $I_2$ and $CH_3I$ under different conditions (static and dynamic adsorption at different temperatures and adsorbate concentrations) for a direct comparison with benchmark adsorbents reported in the literature. Under the static high-concentration conditions, COF-TAPB and COF-TAPT exhibited similar high $I_2$ adsorption capacities, and their static $I_2$ uptake values (7.94 and 8.61 g g$^{-1}$, respectively) are among the highest reported for various adsorbents. Unlike the case of $I_2$ adsorption, COF-TAPT exhibited a significantly higher $CH_3I$ uptake capacity than COF-TAPB, suggesting that the N content in the adsorbent plays a vital role in $CH_3I$ adsorption. The further systematic analysis confirmed that the $CH_3I$ adsorption capacity is positively correlated with the N content in the adsorbent. Remarkably, COF-TAPT exhibits a record-high $CH_3I$ adsorption capacity (1.53 g g$^{-1}$; static conditions at 25 °C), which can be attributed to the combined effect of its high N content (16.1 wt%) and large surface area (~2300 m$^2$ g$^{-1}$). In the dynamic $CH_3I$ adsorption measurement, COF-TAPT demonstrated the highest capacity of all tested adsorbents. When used to simultaneously capture low-concentration $I_2$ (150 ppm) and $CH_3I$ (50 ppm) from a carrier gas stream, COF-TAPT outperformed all tested adsorbents except an ionic COF in terms of total iodine capture. The adsorbed $I_2$ and $CH_3I$ can be easily extracted by ethanol or acetone from COF-TAPT to fully restore its adsorption capacity for subsequent adsorption cycles. The density functional theory (DFT) calculations revealed that the $CH_3I$ binding energy at different N sites in COF-TAPT follows the order imine N > triazine N > $sp^3$ N.

## Results

**Characterization.** Powder X-ray diffraction (PXRD) indicated that COF-TAPT and COF-TAPB are highly crystalline, both exhibiting four clearly discernible peaks before 12° (2θ) and two weak peaks at around 16° and 25° (Fig. 2a, c). We performed structural modeling based on PXRD to corroborate that the synthesized COFs have the designed structures. The results revealed that for both materials, the experimental data agree well with the simulated data based on the eclipsed (AA) stacking model (Fig. 2b, d), and the observed diffraction peaks can be indexed as (100), (110), (200), (210), (310), and (001) reflections, respectively. The Pawley fitting results were reasonably good, producing a unit cell of $a = b = 23.51$ Å, $c = 3.64$ Å, $\alpha = \beta = 90°$, and $\gamma = 120°$ ($R_P = 2.15\%$ and $R_{WP} = 2.89\%$) for COF-TAPT (Supplementary Table 1), and a unit cell of $a = b = 23.61$ Å, $c = 3.3.65$ Å, $\alpha = \beta = 90°$, and $\gamma = 120°$ ($R_P = 2.57\%$ and $R_{WP} = 3.35\%$) for COF-TAPB (Supplementary Table 2). From the $N_2$ sorption isotherms collected at 77 K (Fig. 2e), the Brunauer−Emmett−Teller (BET) surface areas of COF-TAPT and COF-TAPB are derived at 2348 and 2290 m$^2$ g$^{-1}$, respectively. Their pore size distribution is centered at 1.92 nm (Fig. 2e), which is consistent with the designed structure and PXRD results. The two COFs exhibit very similar PXRD patterns and $N_2$ adsorption isotherms, indicating that they have comparable structural and textural properties (Supplementary Table 3). High-resolution transmission electron microscopy confirmed that they both possess a hexagonal structure containing one-dimensional channels (Supplementary Fig. 1).

The completion of the Schiff base reaction between TFPA and TAPT/TAPB was evidenced by the disappearances of bands at 3359 to 3428 cm$^{-1}$ (amino groups) and 1685 cm$^{-1}$ (aldehyde groups) and the synchronous appearance of the characteristic -C=N- stretching band at ~1625 cm$^{-1}$ in the Fourier transform infrared (FTIR) spectra (Fig. 2f). The successful construction of the designed COF frameworks and the presence of different N species in the frameworks were further confirmed by solid-state $^{13}$C nuclear magnetic resonance (NMR) spectroscopy and N 1s X-ray photoelectron spectroscopy (XPS) (Supplementary Fig. 2). In addition, the elemental analysis revealed that the carbon (C), hydrogen (H), and N content in COF-TAPT and COF-TAPB closely agreed with the theoretical values (Supplementary Table 4). These COFs retained high crystallinity after treatment with concentrated $HNO_3$ aqueous solution (5 M) or β-irradiation (200 kGy), exhibiting the excellent stability required to capture radioactive iodine from the off-gas stream (Supplementary Fig. 3).

**Static $I_2$ and $CH_3I$ adsorption.** In most previous studies, $I_2$ adsorption was performed in a static closed system with saturated $I_2$ vapor at 75 °C, and the adsorption capacity was determined based on the mass increase subsequently measured under ambient conditions at room temperature[29,32,33,35–37,56]. To directly compare with the previously developed adsorbents, we evaluated COF-TAPB and COF-TAPT using the same experimental setup (see the Experimental section in the Supporting Information for detailed methods). The results revealed that COF-TAPB and COF-TAPT adsorbed 7.94 and 8.61 g g$^{-1}$ $I_2$ within 96 h, respectively (Fig. 3a and Supplementary Table 3). These values rank high among all adsorbents tested under the same conditions (Fig. 3c and Supplementary Table 5). It is worth noting that under this commonly used evaluation condition, where the concentration of $I_2$ is rather high (~16,000 ppm), the adsorption is dominated by the intermolecular interactions of $I_2$; consequently, the capacity is largely determined by the pore volume of the adsorbent in addition to the characteristics of the binding sites.

We used the average adsorption rate determined at 80% of the full adsorption capacity ($K_{80\%}$)[38] to describe the adsorption kinetics of the adsorbents. Despite their similar porous structures, COF-TAPT exhibited a higher $K_{80\%}$ value than COF-TAPB (0.48 vs. 0.33 g g$^{-1}$ h$^{-1}$), which can be attributed to its higher N content promoting the initial adsorption of $I_2$. The measured $I_2$ adsorption kinetics of COF-TAPT and COF-TAPB are faster than those of many previously reported microporous adsorbents (Supplementary Table 5) due to their highly crystalline structures that facilitate mass transport. To investigate the effect of porosity on the adsorption capacity and adsorption kinetics, we prepared a control sample, which was synthesized using the same process as COF-TAPT, except that $N,N$−dimethylformamide (DMF) instead of mixed ethanol and $o$-dichlorobenzene was used as the solvent. The obtained material (denoted as TFPA-TAPT) has the same chemical composition as COF-TAPT but a much lower porosity (BET specific surface area: 1284 m$^2$ g$^{-1}$; total pore volume: 0.19 cm$^3$ g$^{-1}$) due to its poor crystallinity (Fig. 4a, b). Under the same conditions, TFPA-TAPT exhibited a lower $I_2$ adsorption capacity (4.31 g g$^{-1}$) and a slower adsorption rate ($K_{80\%}$: 0.098 g g$^{-1}$ h$^{-1}$) than COF-TAPT (Supplementary Fig. 4c). This result demonstrates the significant influence of textural properties of the adsorbent on its $I_2$ adsorption behavior.

We also evaluated the $CH_3I$ adsorption performance of COF-TAPT and COF-TAPB in a static closed system with a saturated $CH_3I$ vapor at 75 °C, as used in the previous study[34], for direct comparison. The results demonstrated that COF-TAPT adsorbed 1.53 g g$^{-1}$ of $CH_3I$ (Fig. 3b), exceeding the capacity of the state-of-the-art $CH_3I$ adsorbent SCU-COF-2 (1.45 g g$^{-1}$)[34]. In contrast, COF-TAPB adsorbed only 0.81 g g$^{-1}$ of $CH_3I$ under the same

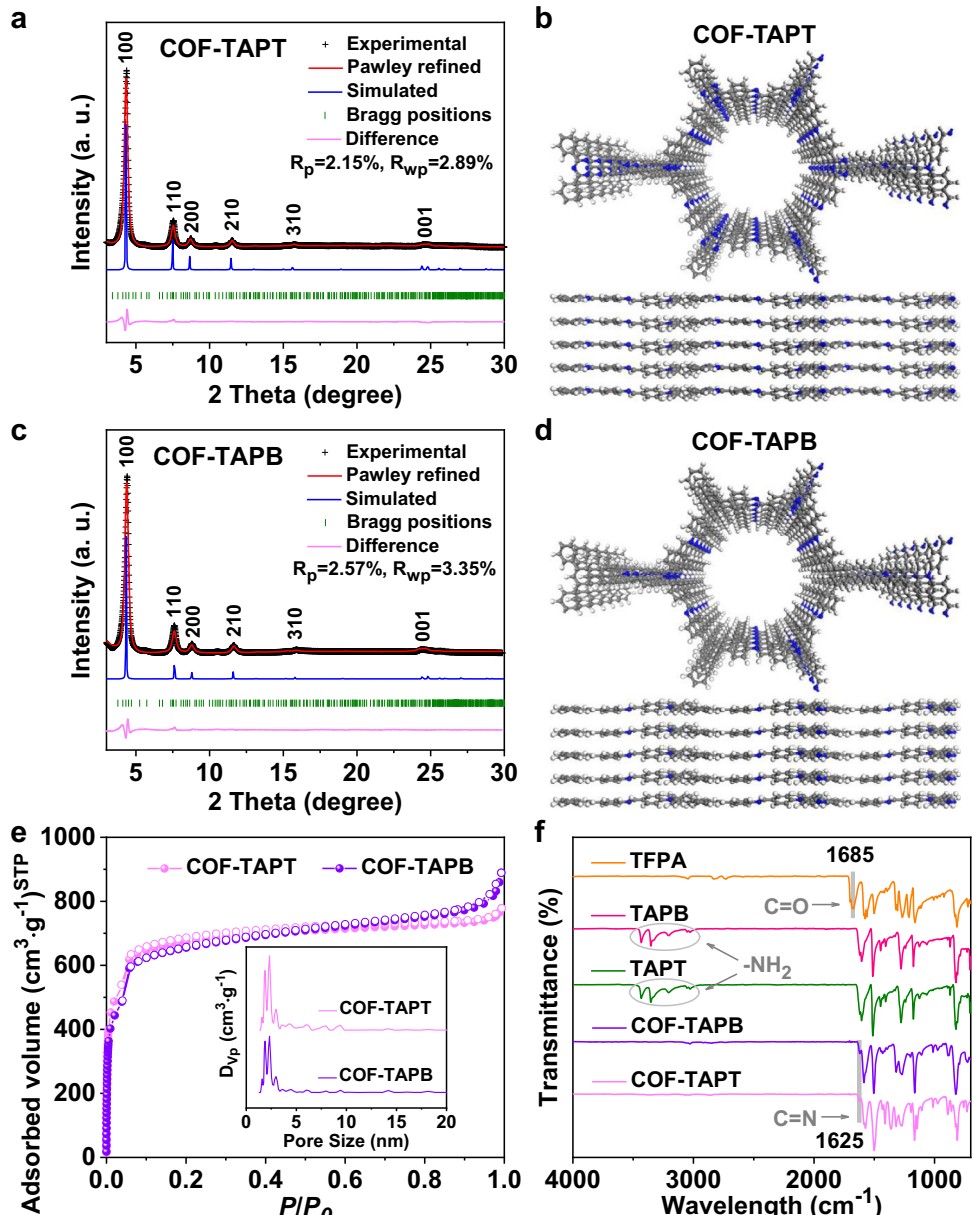

**Fig. 2 Structural characterization of COF-TAPT and COF-TAPB. a, c** PXRD patterns of **a** COF-TAPT and **c** COF-TAPB, with the experimental profiles in black, Pawley refined profiles in red, calculated (from the refined structure model) profiles in blue, and differences between experimental and refined PXRD patterns in pink. Green lines indicate Bragg positions. **b, d** Refined structure model of **b** COF-TAPT and **d** COF-TAPB viewed along the c-axis (upper) and a-axis (lower). **e** $N_2$ sorption isotherms of COF-TAPT and COF-TAPB. Inset shows the derived pore size distribution profiles of COF-TAPT and COF-TAPB. **f** FTIR spectra of the synthetic precursors (TFPA, TAPB, and TAPT) and the two COFs (COF-TAPB and COF-TAPT).

conditions (Fig. 3b). The control sample TFPA-TAPT exhibited a $CH_3I$ adsorption capacity of 1.37 g g$^{-1}$, although its BET surface area is only half of that of COF-TAPT (Supplementary Fig. 4d). These results collectively indicate that, unlike $I_2$ adsorption, $CH_3I$ adsorption capacity depends on the number of strong binding sites of the adsorbent rather than its surface area or pore volume. This conclusion was further verified by the plots of $CH_3I$ uptake vs. the BET surface area (Supplementary Fig. 5a) and $CH_3I$ uptake vs. N content (Supplementary Fig. 5b), based on the data for seven adsorbents, which clearly indicate that the $CH_3I$ uptake is irrelevant to the surface area but positively correlated with the N content of the adsorbent (Supplementary Table 3). In addition, at their full adsorption capacity, the $CH_3I/N$ ratios in COF-TAPT and COF-TAPB were 0.97 and 0.89, respectively, whereas the $I_2/N$ molar ratios were 3.05 and 4.89, respectively.

These results collectively indicate the essential difference between $I_2$ adsorption and $CH_3I$ adsorption. The nearly one-to-one correspondence between $CH_3I$ and N suggests that $CH_3I$ molecules are only adsorbed on N sites, presumably by forming salts. In contrast, $I_2$ adsorption can occur at other sites in the π-conjugated frameworks (e.g., benzene rings) or be promoted by forming polyiodide species, resulting in high $I_2/N$ ratios.

**Dynamic $I_2$ and $CH_3I$ adsorption.** Given the low concentrations of molecular iodine and organic iodides (usually <200 ppm) in the off-gas stream[2,57,58], it is more meaningful to test the adsorption behavior of adsorbents at low $I_2$ (or $CH_3I$) concentrations related to practical applications. To explore the potential of the developed COFs for practical applications, we evaluated their $I_2$ and $CH_3I$ adsorption capacities under dynamic

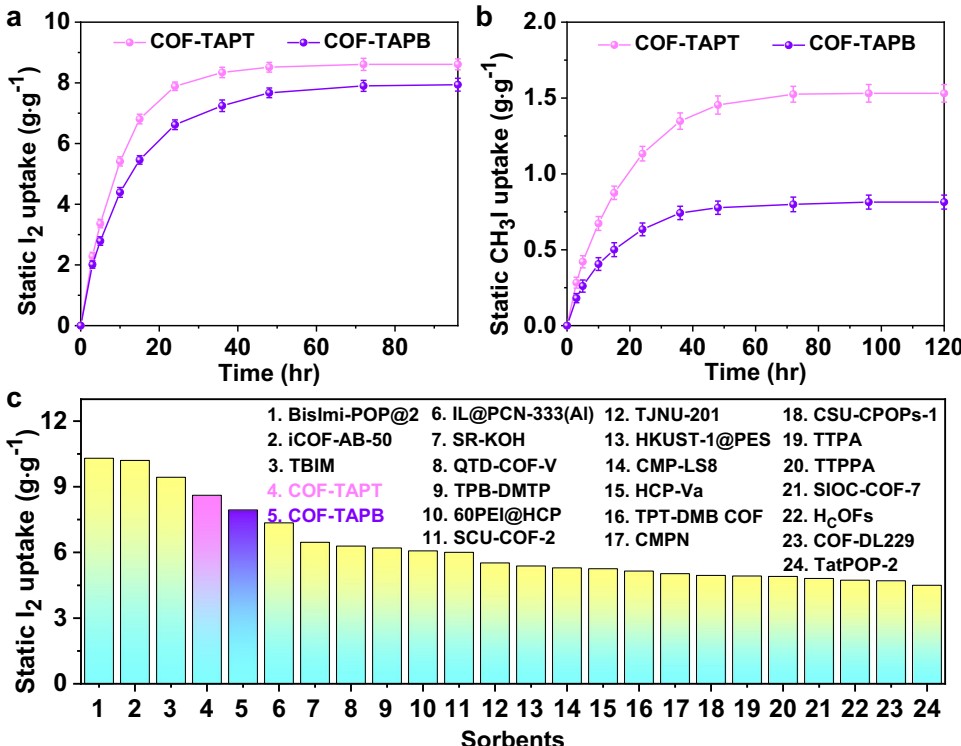

**Fig. 3 Static I₂ and CH₃I adsorption performance.** Gravimetric measurement of static **a** I₂ and **b** CH₃I vapor adsorption capacities of COF-TAPT and COF-TAPB materials as a function of time at 75 °C. **c** Comparison of the static I₂ adsorption capacities of various high-performance adsorbents. The specific I₂ uptake values of the reported adsorbents and corresponding references are presented in Supplementary Table 5. The error bars are the standard deviations from three parallel measurements.

conditions using a fixed-bed column-breakthrough configuration, which allowed the concentration of I₂ and CH₃I, the temperature of the adsorbent bed, and humidity to be freely adjusted[42].

To directly compare the developed COFs with benchmark adsorbents[17,34,42], we conducted dynamic adsorption at 25 °C with 400 ppm of I₂ in N₂ flow (adsorbent: 30.0 mg; flow rate: 10.0 mL min⁻¹). Under these conditions, COF-TAPB exhibited a steep breakthrough step after 45 h with a total I₂ uptake of 2.18 g g⁻¹, and COF-TAPT demonstrated a similar breakthrough profile, whereas its breakthrough time was 50 h, corresponding to a total I₂ uptake of 2.38 g g⁻¹ (Fig. 4a). The observed I₂ adsorption capacities are significantly higher than those of most reported adsorbents under similar measurement conditions (Supplementary Fig. 6 and Supplementary Table 6). In addition, the presence of water vapor only caused a slight decrease in the I₂ uptake of COF-TAPT (from 2.38 g g⁻¹ to 2.32 g g⁻¹) and COF-TAPB (from 2.18 to 2.13 g g⁻¹) (Fig. 4b), indicating that they are both tolerant to moisture, which is important for practical applications.

We performed dynamic CH₃I adsorption using the same column-breakthrough setup, with the CH₃I concentration controlled at 200,000 ppm. The results revealed that, under dry conditions, the CH₃I uptakes of COF-TAPB and COF-TAPT were 0.71 and 1.30 g g⁻¹, respectively (Fig. 4c, d). The CH₃I adsorption capacity of COF-TAPT (1.30 g g⁻¹) is higher than that of all reported adsorbents except MIL-101-Cr-HMTA[1] (Supplementary Fig. 7 and Supplementary Table 7). The ultrahigh CH₃I adsorption capacity of MIL-101-Cr-HMTA is attributed to the combination of a large surface area and abundant tertiary amine groups that can strongly and specifically interact with CH₃I. However, MIL-101-Cr-HMTA exhibits a limited adsorption capacity for I₂ (Supplementary Fig. 8 and Supplementary Table 6) due to the lack of effective I₂ binding sites other than tertiary amines. When water

vapor was introduced into the feed stream (relative humidity = 50%), the CH₃I adsorption capacity of COF-TAPT decreased from 1.30 to 1.06 g g⁻¹, indicating competitive adsorption between H₂O and CH₃I. This result is because the adsorption of CH₃I primarily occurs on N sites, which also adsorb H₂O molecules. The dynamic adsorption measurements indicate that COF-TAPT has a similar I₂ uptake capacity but a significantly higher CH₃I uptake capacity than COF-TAPB. This outcome is consistent with the results of the static adsorption experiments and further validates the above conclusion about the difference between I₂ adsorption and CH₃I adsorption.

The adsorbed I₂ and CH₃I in COF-TAPT can be fully extracted by ethanol to regenerate its adsorption capacities. When the I₂-saturated COF-TAPT (I₂@COF-TAPT) or CH₃I-saturated COF-TAPT (CH₃I@COF-TAPT) was immersed in ethanol, the I₂ or CH₃I desorption process proceeded spontaneously and accelerated with the assistance of sonication. As a commonly used method for regenerating adsorbents after I₂ adsorption, extraction with ethanol can also efficiently remove CH₃I from COF-TAPT because the N-methylation reaction is reversible in protic solvents[59]. The regenerated COF-TAPT (COF-TAPT-Re) restored its original physiochemical properties, as evidenced by the FTIR, PXRD, and N₂ sorption characterizations. Correspondingly, the exceptionally high adsorption capacity of COF-TAPT can be fully restored in four successive adsorption/extraction cycles under the above conditions (Supplementary Figs. 9, 10).

**Simultaneous capture of low-concentration I₂ and CH₃I.** After evaluating the adsorption performance of the developed COFs for I₂ and CH₃I under commonly used conditions, we explored their ability to capture I₂ and CH₃I at much lower concentrations (150 and 50 ppm, respectively) relevant to practical off-gas treatment applications. Several state-of-the-art adsorbents for I₂ or CH₃I

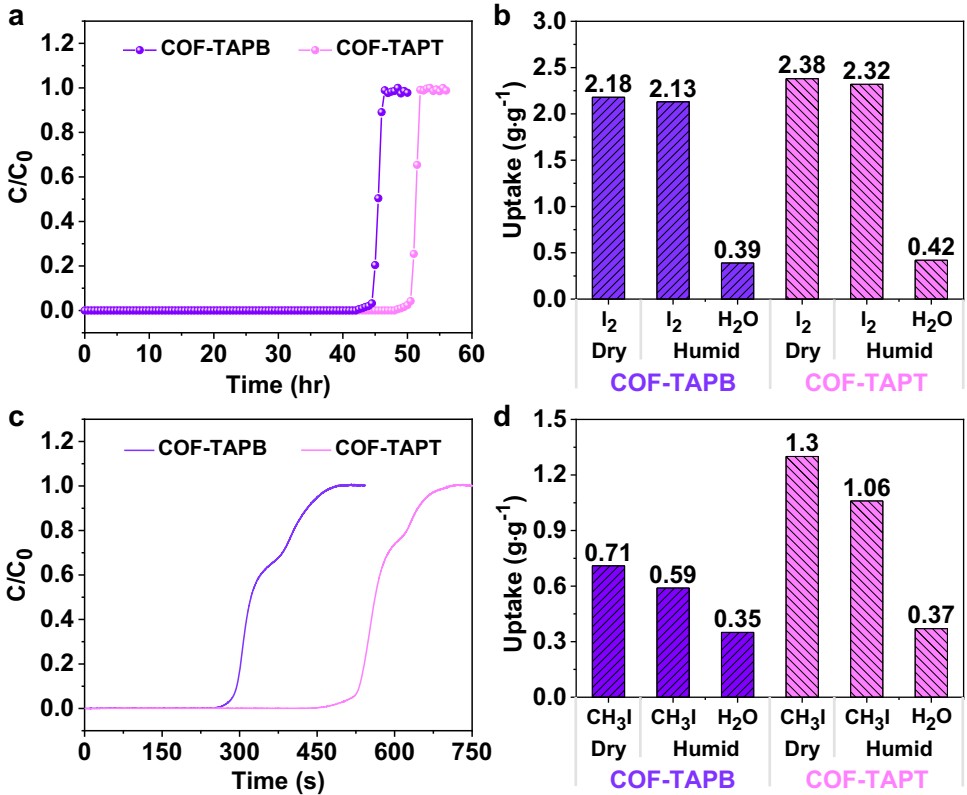

**Fig. 4 Dynamic I$_2$ adsorption and CH$_3$I adsorption performances. a** I$_2$ breakthrough profiles of COF-TAPB and COF-TAPT at 25 °C. The concentration of I$_2$ in the carrier gas is 400 ppm. **b** Dynamic I$_2$ uptake values of COF-TAPB and COF-TAPT under dry and humid conditions. The water uptake under humidity is also presented. **c** CH$_3$I breakthrough profiles of COF-TAPB and COF-TAPT at 25 °C. The concentration of CH$_3$I in the carrier gas is 200,000 ppm. **d** Dynamic CH$_3$I uptake values of COF-TAPB and COF-TAPT under dry and humid conditions. The water uptake under humidity is also presented.

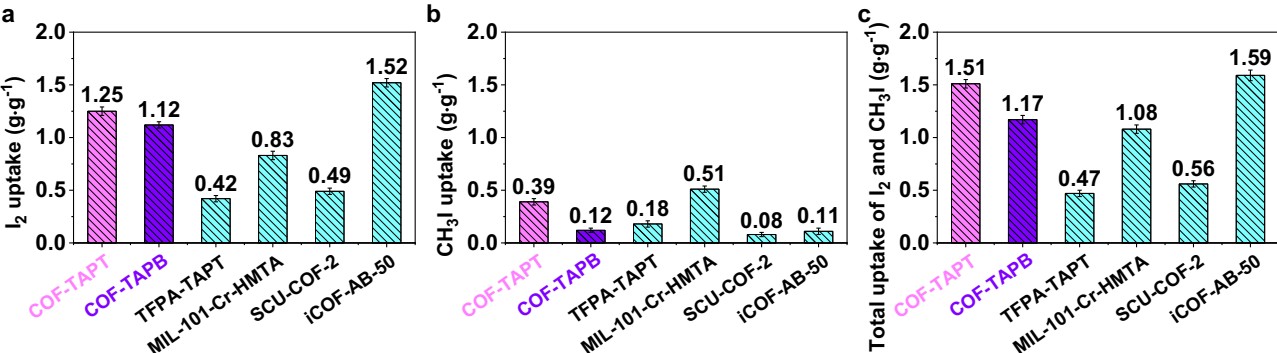

**Fig. 5 Comparison of different adsorbents in dynamic adsorption capacity.** The capacities were measured from **a** single-component I$_2$ at 150 ppm, **b** single-component CH$_3$I at 50 ppm, **c** mixed-component (150 ppm I$_2$ and 50 ppm CH$_3$I) breakthrough experiments performed at 25 °C. The error bars are the standard deviations from three parallel measurements.

capture, including MIL-101-Cr-HMTA[1], SCU-COF-2[34], and iCOF-AB-50[42], were evaluated under the same conditions for comparison. We started with single-component measurements, introducing only I$_2$ (150 ppm) or CH$_3$I (50 ppm) in the feed stream for capture. The results revealed that, for I$_2$ capture, the order of adsorption capacity of the tested adsorbents is iCOF-AB-50 (1.52 g g$^{-1}$) > COF-TAPT (1.25 g g$^{-1}$) > COF-TAPB (1.12 g g$^{-1}$) > MIL-101-Cr-HMTA (0.83 g g$^{-1}$) > SCU-COF-2 (0.49 g g$^{-1}$) > TFPA-TAPT (0.42 g g$^{-1}$) (Fig. 5a and Supplementary Table 8). This order is consistent with that derived from the static adsorption measurements (Fig. 3c). The exceptionally high I$_2$ uptake of iCOF-AB-50 is attributed to the presence of abundant ionic groups that

effectively promote I$_2$ adsorption via strong Coulomb interactions. The results of various adsorbents capturing CH$_3$I, as summarized in Fig. 5b and Supplementary Table 8, indicate that COF-TAPT ranks second among all tested adsorbents. The specific CH$_3$I adsorption capacities are as follows: MIL-101-Cr-HMTA (0.51 g g$^{-1}$) > COF-TAPT (0.39 g g$^{-1}$) > TFPA-TAPT (0.18 g g$^{-1}$) > COF-TAPB ≈ iCOF-AB-50 (0.12 g g$^{-1}$) > SCU-COF-2 (0.08 g g$^{-1}$).

These results further confirm that the adsorption behaviors of I$_2$ and CH$_3$I are different. The adsorption of I$_2$ can be initiated through specific functional groups (e.g., ionic groups) in the adsorbent and further promoted through strong intermolecular interactions. Therefore, the type of binding sites and textural

properties of the adsorbent both critically influence the $I_2$ uptake. In contrast, the adsorption of $CH_3I$ is primarily determined by the type and number of binding sites and is not much related to the textural properties of the adsorbent. In addition, ionic groups can strongly promote the adsorption of $I_2$ but have a little promotional effect on the adsorption of $CH_3I$ because $I_2$ can be easily induced to form charge species, such as $I_3^-$ and $I_5^-$, whereas $CH_3I$ cannot.

Finally, we measured the simultaneous capture of iodine species on various adsorbents by co-feeding $I_2$ (150 ppm) and $CH_3I$ (50 ppm). Considering the excellent capture ability of COF-TAPT for $I_2$ and $CH_3I$ in the single-component measurements, good performance in the co-capture of these two species is expected. Indeed, COF-TAPT exhibited a significantly higher total iodine ($I_2 + CH_3I$) capture capacity (1.51 g g$^{-1}$) than other tested adsorbents, except iCOF-AB-50 (Fig. 5c and Supplementary Table 8). The total uptake of iCOF-AB-50 primarily derives from the contribution of $I_2$ adsorption; thus, it is conceivable that COF-TAPT is more suitable for feed streams containing a high fraction of organic iodide. Similar to the single-component measurement results, the high performance of the COF-TAPT in simultaneous capture of $I_2$ and $CH_3I$ can be fully regenerated in successive tests (Supplementary Fig. 11).

## Discussion

To analyze the adsorption sites of COF-TAPT, we characterized the $I_2$-saturated COF-TAPT sample ($I_2$@COF-TAPT) with PXRD after the static adsorption measurement. The obtained PXRD pattern did not exhibit diffraction peaks related to the original crystalline structure of COF-TAPT (Supplementary Fig. 9b), indicating the loss of structural order due to the incorporation of $I_2$ into the porous channels. Moreover, no diffraction peaks associated with $I_2$ were observed, ruling out the possibility that the high $I_2$ adsorption capacity of COF-TAPT was caused by the recrystallization of $I_2$ outside its porous structure. Solid-state $^{13}$C NMR spectra revealed that the chemical shifts of all carbon atoms in COF-TAPT changed to a certain extent after the adsorption of $I_2$ (Fig. 6a), suggesting that $I_2$ molecules interacted with various sites throughout the entire π-electron conjugated framework[27,34]. The N 1$s$ XPS indicated that after the adsorption of $I_2$, the peak fractions at 398.6 and 399.5 eV, assigned to imine/triazine N and $sp^3$ N in COF-TAPT[33,34], shifted to 400.7 and 401.4 eV, respectively (Fig. 6b), suggesting the formation of charge-transfer complexes between $I_2$ and various N species in COF-TAPT. In the FTIR spectra, the adsorption of $I_2$ caused the bands of C=N at 1625 cm$^{-1}$, C=C at 1586 cm$^{-1}$, C–N (in N-ph$_3$) at 1361 cm$^{-1}$, and C–N (in ph-N=C) at 1195 cm$^{-1}$ to decrease in intensity or shift (Fig. 6c), indicating that all functional groups in the entire framework of COF-TAPT interact with $I_2$[27,33,34]. The Raman spectrum of $I_2$@COF-TAPT exhibited characteristic bands at 107.5, 142.2, and 166.7 cm$^{-1}$, which can be assigned to the symmetric stretching vibration of $I_3^-$, asymmetric stretching vibration of $I_3^-$, and stretching vibration of $I_5^-$, respectively (Fig. 6d)[60,61]. These spectroscopic observations indicate that various functional groups in COF-TAPT, including phenyl rings, imine and triazine moieties, and $sp^3$ N, were all involved in forming the charge-transfer complex with $I_2$ and that polyiodide species were produced during the adsorption process.

The $CH_3I$-saturated COF-TAPT sample ($CH_3I$@COF-TAPT) was also characterized using $^{13}$C NMR, N 1$s$ XPS, and FTIR. Compared to the original material, $CH_3I$@COF-TAPT exhibited an intense new signal at ~58.9 ppm in the $^{13}$C NMR spectrum, originating from $CH_3I$ that formed salts at various N sites

through methylation reactions (Fig. 6a)[34]. In addition, after the adsorption of $CH_3I$, the N 1$s$ electron binding energies of imine/triazine N and $sp^3$ N in COF-TAPT increased by 1.3 and 3.7 eV, respectively, providing additional evidence for the binding of $CH_3I$ on N species (Fig. 6b)[1,34]. Compared with $I_2$ adsorption, $CH_3I$ adsorption resulted in a less pronounced peak shift for imine/triazine N and a more pronounced peak shift for $sp^3$ N, implying differences in affinity of $I_2$ and $CH_3I$ at different N sites. In FTIR, the adsorption of $CH_3I$ on COF-TAPT also resulted in the intensity change or shift of the characteristic bands of C=N and C–N bonds, and the appearance of a band at ~939 cm$^{-1}$ indicated the formation of new C–N bonds on the heterocycles in COF-TAPT (Fig. 6c)[31,34]. Notably, the bands of C=C at 1560–1590 cm$^{-1}$ were unchanged upon the adsorption of $CH_3I$. These spectroscopic results support the conclusion that $CH_3I$ molecules are not adsorbed on the benzene ring moieties in COF-TAPT but are specifically bonded to the nucleophilic N sites through N-methylation reactions. The $CH_3I$-adsorbed COF-TAPT exhibits strong anion-exchange ability (Supplementary Fig. 12), confirming the generation of exchangeable iodide ions.

There are three types of nucleophilic N species (i.e., imine, triazine, and $sp^3$ N) in COF-TAPT, all of which can bind to $CH_3I$. To gain more insight into the preferential binding sites of $CH_3I$, we performed DFT calculations to assess their binding energies with $CH_3I$. The calculations were conducted at the B3LYP[62,63] level of the exchange functional, using TFPA-T as the model molecule to represent COF-TAPT (Fig. 7). The calculated binding energy is −15.0 kcal mol$^{-1}$ for imine, −5.4 kcal mol$^{-1}$ for triazine, and −2.6 kcal mol$^{-1}$ for $sp^3$ N sites, indicating that the imine groups in the COF are the most favorable adsorption sites for $CH_3I$ (Fig. 7). This result suggests that introducing imine groups and maximizing their content in the adsorbent may be a good choice to improve the ability to capture low-concentration $CH_3I$. In previous studies, $sp^3$ N promoted the capture of $CH_3I$, where N was connected with alkyl chains[1,47,48]. However, in TFPA-T, $sp^3$ N is connected with three benzene rings; thus, its nucleophilicity is greatly reduced due to the conjugation effect. We note that the XPS data (Fig. 6b) suggest that $CH_3I$ interacts more strongly with $sp^3$ N than with imine/triazine N (more pronounced peak shift). This discrepancy is not fully understood and requires further exploration.

In conclusion, the development of high-performance adsorbents for the simultaneous capture of molecular iodine and organic iodides relies on understanding the similarities and differences between these two processes. Based on previous studies, we hypothesized that N-rich carbonaceous adsorbents are conducive to simultaneously capturing $I_2$ and $CH_3I$. Further studies revealed that $I_2$ could be relatively easily adsorbed on a variety of electron-donor sites, including various N species and aromatic moieties, by forming charge-transfer complexes and polyiodides. Therefore, the characteristics of binding sites and textural properties (e.g., surface area and pore volume) of the adsorbent both affect $I_2$ uptake. The adsorption of $CH_3I$ occurs specifically on nucleophilic N sites through N-methylation reactions to form salts and is unrelated to the textural properties of the adsorbent. In addition, ionic groups can strongly promote the adsorption of $I_2$ but have little promotional effect on the adsorption of $CH_3I$. These findings motivated the development of a COF-based adsorbent, COF-TAPT, which combines a high surface area and numerous nucleophilic N sites, including imine, triazine, and $sp^3$ N, thereby exhibiting excellent adsorption capacity for $I_2$ and $CH_3I$. We evaluated COF-TAPT for $I_2$ and $CH_3I$ adsorption under different conditions and found that it outperforms most state-of-the-art adsorbents in all measurements, especially at low-concentration conditions relevant to practical applications. In addition, we calculated the binding energies of $CH_3I$ at different

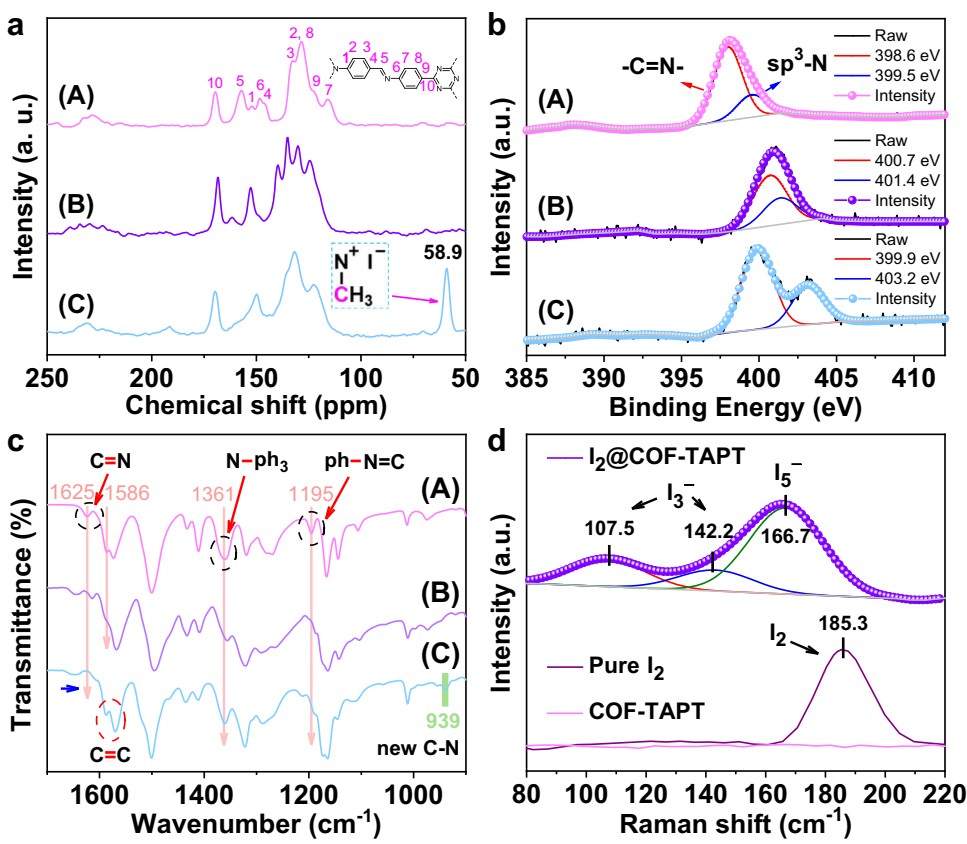

**Fig. 6 Characterization of I₂/adsorbent and CH₃I/adsorbent interactions. a** ¹³C NMR spectra of (A) pristine, (B) I₂-saturated, and (C) CH₃I-saturated COF-TAPT. **b** N 1s XPS spectra of (A) pristine, (B) I₂-saturated, and (C) CH₃I-saturated COF-TAPT. **c** FTIR spectra of (A) pristine, (B) I₂-saturated, and (C) CH₃I-saturated COF-TAPT. **d** Raman spectra of pure I₂, pristine COF-TAPT, and I₂-saturated COF-TAPT.

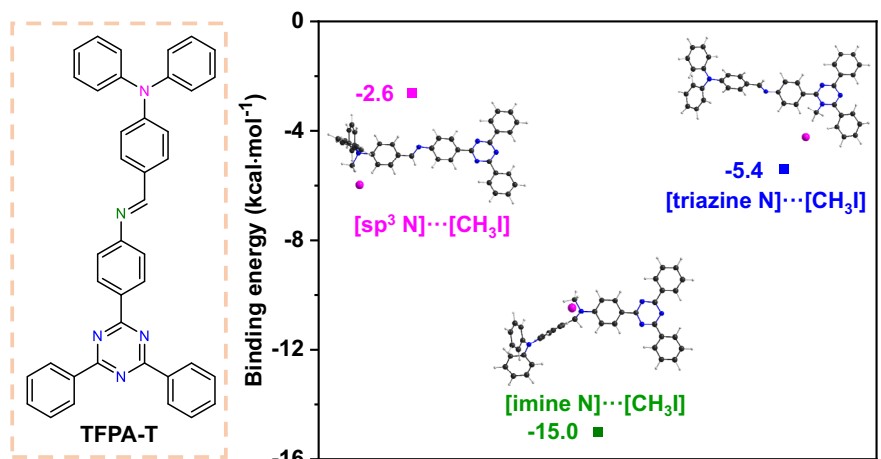

**Fig. 7 Density functional theory calculations of the binding energies of CH₃I with different N sites.** TFPA-T is the model molecule used for calculations that represent COF-TAPT. In the molecular structure of TFPA-T (left panel), $sp^3$ N, imine N, and triazine N are labeled in pink, green, and blue, respectively.

N sites, and the results revealed that imine groups might be the most preferred adsorption sites.

## Data availability

All data supporting the findings of this study are available within the article and the Supplementary information file, or available from the corresponding authors on request.

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

## Acknowledgements

This research is supported by the AMPM CCF fund (FCC/1/1972-43-01) to Y.H. from King Abdullah University of Science and Technology.

## Author contributions

Y.X. and Y.H. conceived the project and designed the experiments. Y.X. and W.A.M. prepared and characterized the materials. Y.X., T.P., and X.D. conducted the adsorption experiments. C.C. performed TEM measurement and analysis. Y.X. and Y.Y. performed PXRD simulations and analysis. L.Z. helped the irradiation treatments. L.C. performed the DFT calculations. Y.X., Q.L., I.P., and Y.H. wrote the manuscript with contributions from all the authors.

## Competing interests

The authors declare no competing interests.
