## [Peer Review File · Nature Communications]

Efficient and Simultaneous Capture of Iodine and Methyl Iodide Achieved by a Covalent Organic FrameworkReviewers' Comments:

Reviewer #1:

Remarks to the Author:

In this manuscript, two COFs (COF-TAPT and COF-TAPB) was synthesized for the simultaneous capture of iodine and organic iodides at low concentrations. I believe this work is suitable to be published in Nat. Commun. after addressing the following comments:

1. For the intro- part, the authors are supposed to encapsulate in detail about the different iodine/iodide species and isotopes in the off-gas to reinforce the scientific significance of this work.
2. The authors claim their material is mainly based on the filling of iodine molecules into the pores and the enhanced host-guest interaction. Please calculate the estimated adsorption capacity according to the pore volume of COF-TAPT and COF-TAPB as the previous works did (Jiang et al. Adv. Mater. 2018, 30, 1801991) and compare this value to the experimental value to further support their claim.
3. In the part of material regeneration, the I₂- or CH₃I- loaded COF was immersed into ethanol or acetone, and ultrasonicated for a certain period to obtain regenerated COF. Since the captured I₂ can be dissociation from material due to the relatively weak interactions, I wonder why the CH₃I-loaded material can be fully regenerated at such a mild condition after the methylation? Please specify it.
4. From the XPS results before and after the adsorption (I₂/CH₃I), the peaks assigned to sp³-N are gradually shifted to higher binding energy, indicating the increase of binding energy from relatively weak interaction (between I₂ and N) to stronger interaction (between CH₃I and N). However, the specie of N (-C=N-) present another pattern, which may help reveal the affinity of different N atoms toward different species of iodine (Also, the authors listed the structure activity relationship between the N content and CH₃I uptake). Please supplement quantitative and statistical analyses of the phenomenon for better understanding.
5. Please provide the EDS mapping before and after adsorption (iodine and org-I) to show the distribution of the iodine elements in materials.

Reviewer #2:

Remarks to the Author:

The authors synthesized two COFs to capture radioactive molecular I₂ and CH₃I. The design of the COFs is conventionally based on the consideration of introducing N sites, while the difference of the two COFs is the N content in the framework. Hence, the thought of structure design is not innovative enough. A merit of this work is the simultaneous adsorption of I₂ and CH₃I, the performance of the synthesized COFs is good but not excellent. The static I₂ uptake ranks 4th and 5th compared with other 22 materials. The dynamic adsorption capacity of the synthesized COFs is also not the best. Moreover, the t_{80%} is 18 h and 24 h for the synthesized COFs, which are longer than many of other materials, representing the low adsorption dynamic performance of the two COFs. Therefore, from both the viewpoints of the features of COF structure design and the adsorption performance, this work is not recommended to be published in nature communication. Following questions are suggested to be addressed:

1. Figure 5d, the I₂ transformed into I³⁻ and I⁵⁻, what's the origin of the negative charge. If the COF framework takes one positive charge, will the COF be stable under this condition?
2. The density of solid I₂ is 4.93 g/cm³, the pore volumes of COF-TAPT and COF-TAPB are lower than 1 cm³/g. Why the static uptake is high to 7~8 g/g? Whether the I₂ vapor is condensed to the outside surface of COFs when they are weighed at room temperature?
3. The I₂ and CH₃I are radioactive, stability of the COFs should be tested under radiation condition.

4. The conclusion of "The CH₃I adsorption capacity is positively correlated with the content of strong binding sites but is unrelated to the textural properties of the adsorbent." is one-sided as the number of COFs tested in this work is small.
5. The content is similar to the previously published paper *Angew. Chem. Int. Ed.* 2021, 60, 2 – 11.

Reviewer #3:

Remarks to the Author:

Simultaneous capture of iodine and methyl iodide by porous materials through breakthrough test remains a challenge. As stated in the text, there is only one COF that covers this feature. In this work, Han et al prepared two comparable COFs but with the difference in nitrogen atom, thus, giving an insight into the relationship between I₂, CH₃I uptake and various nitrogen atoms. Effective and simultaneous capture of iodine and methyl iodide were observed for the nitrogen-rich COFs. The results are very interesting. The preparation of this work was careful. But some points should be considered before publication.

1. "To the best of our knowledge, there is only one COF material (SCU-COF-2) evaluated for both I₂ and CH₃I adsorption". In fact, Han and co-workers has also reported a MOF for such use. Thus, please revise this sentence.
2. Although the author has made deep research in both I₂ and CH₃I breakthrough test, however, as investigated in their previous work, simultaneous capture of iodine and methyl iodide at high temperature such as 423 K is suggested to be investigated, especially in the presence of HNO₃.
3. How about the irradiation stability of this material?
4. How about the stability of this material under HNO₃?
5. The pore size is not given in Fig. 1.
6. In Fig. 4, the uptake is based on weighing method, thus, error bar is suggested.
7. The adsorption of anion ion or anionic dye is suggested to further confirm the CH₃I adsorption through methylation reaction.
8. How about the recycle use of simultaneous capture of iodine and methyl iodide at ppm level.

Point-by-Point Responses to the Reviewers' Comments

Reviewer #1

Comments: In this manuscript, two COFs (COF-TAPT and COF-TAPB) was synthesized for the simultaneous capture of iodine and organic iodides at low concentrations. I believe this work is suitable to be published in Nat. Commun. after addressing the following comments.

Response: We are truly grateful to the reviewer for his/her appreciation of our work.

1. For the intro- part, the authors are supposed to encapsulate in detail about the different iodine/iodide species and isotopes in the off-gas to reinforce the scientific significance of this work.

Response: Following this suggestion, we elaborated in the revised manuscript the composition of the off-gas and its adverse effects on the environment and human health.

“One of the major safety issues is the volatile radioactive waste produced during the reprocessing of spent nuclear fuels, which primarily consists of radionuclides, such as ^{129}I and ^{131}I in the form of molecular iodine (I_2) or organic iodides (e.g., methyl iodide (CH_3I) and ethyl iodide).¹⁻⁵ These compounds are harmful to the environment (^{129}I has an extremely long half-life of approximately 1.57×10^7 years) or severely affect human metabolism by damaging the thyroid gland, and must be removed before the off-gas is discharged.⁶⁻⁸”

2. The authors claim their material is mainly based on the filling of iodine molecules into the pores and the enhanced host-guest interaction. Please calculate the estimated adsorption capacity according to the pore volume of COF-TAPT and COF-TAPB as the previous works did (Jiang et al. Adv. Mater. 2018, 30, 1801991) and compare this value to the experimental value to further support their claim.

Response: We thank the reviewer for this insightful comment. In many previous studies, I_2 adsorption was performed in a static closed system with saturated I_2 vapor at 75°C , and the adsorption capacity was determined based on the mass increase subsequently measured under ambient conditions. The I_2 adsorption capacity determined in this way is often higher than the theoretical value calculated from the pore volume and the density of solid iodine (see Fig. R1 for summary data). A plausible explanation for this phenomenon is that under such measurement

conditions, there is a large amount of I₂ adsorbed at the external surface of the adsorbent particles and condensed in the inter-particle pores. In this sense, the static I₂ uptake value measured in this way is not a meaningful criterion for performance evaluation. We will discuss this issue shortly in a separation publication.

In the current study, we performed this measurement only for comparison purpose, whereas we highlighted the ability of COF-TAPT to simultaneously capture low-concentration I₂ and CH₃I under dynamic adsorption conditions.

Fig. R1 Static I₂ adsorption capacities and pore volumes of various reported adsorbents. The straight line, plotted by the equation of $q_T = (\text{pore volume} \times 4.93) \text{ g.g}^{-1}$, represents theoretical adsorption capacities (q_T) determined based on the pore volumes (cm³/g) and the density of solid iodine (4.93 g/cm³). This summary indicates that many adsorbents (those above the straight line) exhibit static I₂ adsorption capacities higher than theoretical values.

3. In the part of material regeneration, the I₂- or CH₃I- loaded COF was immersed into ethanol or acetone, and ultrasonicated for a certain period to obtain regenerated COF. Since the captured I₂ can be dissociation from material due to the relatively weak interactions, I wonder why the CH₃I-loaded material can be fully regenerated at such a mild condition after the methylation? Please specify it.

Response: We thank the reviewer for raising this question. Yes, the captured CH₃I can be released by the ultra-sonication-assisted extraction process using ethanol. Taking the imine N (the most favorable binding sites for CH₃I in COF-TAPT according to DFT calculations (Fig. 6)) as an example, its

methylation to form an iminium salt is a reversible reaction in protic solvents (e.g., H₂O, alcohols).⁹ In addition, ultra-sonication treatment, which is a common method for promoting the decomposition of organic molecules,¹⁰⁻¹³ can effectively facilitate this reversible reaction.

In the revised manuscript, we discuss this point following the reviewer's suggestion.

“...and accelerated with the assistance of sonication. As a commonly used method for regenerating adsorbents after I₂ adsorption, extraction with ethanol can also efficiently remove CH₃I from COF-TAPT because the N-methylation reaction is reversible in protic solvents. The regenerated COF-TAPT...”

4. From the XPS results before and after the adsorption (I₂/CH₃I), the peaks assigned to sp³-N are gradually shifted to higher binding energy, indicating the increase of binding energy from relatively weak interaction (between I₂ and N) to stronger interaction (between CH₃I and N). However, the specie of N (-C=N-) present another pattern, which may help reveal the affinity of different N atoms toward different species of iodine (Also, the authors listed the structure activity relationship between the N content and CH₃I uptake). Please supplement quantitative and statistical analyses of the phenomenon for better understanding.

Response: We thank the reviewer very much for his/her valuable insight and suggestion. Following the reviewer's suggestion, we briefly discuss in the revised manuscript the XPS results that imply the different affinity of I₂ and CH₃I at different N sites.

“...at various N sites through methylation reactions (Fig. 5a). In addition, after the adsorption of CH₃I, the N 1s electron binding energies of imine/triazine N and sp³ N in COF-TAPT increased by 1.3 and 3.7 eV, respectively, providing additional evidence for the binding of CH₃I on N species (Fig. 5b). Compared with I₂ adsorption, CH₃I adsorption resulted in a less pronounced peak shift for imine/triazine N and a more pronounced peak shift for sp³ N, implying differences in affinity between I₂ and CH₃I at different N sites. In FTIR...”

To reinforce the established correlation between the CH₃I uptake and the N content of the adsorbent, we include two additional COFs materials (i.e., COF-OH-0¹⁴ and TPB-DMTP-COF¹⁵) in the analysis. The results for seven adsorbents yield a good linear relationship between the CH₃I uptake (at 75 °C) and the N content of the adsorbent with a reasonable coefficient of determination (R²=0.904). In the revised manuscript, these additional data and statistical analysis results are shown in Fig. S5, according to the reviewer's suggestion.

Fig. S5 Relationships between static CH₃I uptakes at 75 °C of different samples and their (a) BET surface areas, and (b) N contents. The grey line in (b) represents the relationship between theoretical CH₃I uptake and N content assuming that each N atom interacts with one CH₃I molecule. The red line in (b) is the linear fitting result based on the seven data points, giving a reasonably good coefficient of determination (R^2) of 0.904.

5. Please provide the EDS mapping before and after adsorption (iodine and org-I) to show the distribution of the iodine elements in materials.

Response: We thank the review for this suggestion. When the I₂/CH₃I loading is high (static adsorption at high concentrations), EDS mapping conducted during SEM imaging can reveal the distribution of iodine in the adsorbent. Please refer to the Fig. R2 below. When the I₂/CH₃I loading is low (dynamic adsorption at low concentrations), EDS mapping data is too noise to show useful information. Therefore, we show the results here for review purpose only.

Fig. R2 SEM image and EDS element mapping of (a) I₂@COF-TAPT and (b) CH₃I@COF-TAPT.

Reviewer #2

Comments: The authors synthesized two COFs to capture radioactive molecular I₂ and CH₃I. The design of the COFs is conventionally based on the consideration of introducing N sites, while the difference of the two COFs is the N content in the framework. Hence, the thought of structure design is not innovative enough. A merit of this work is the simultaneous adsorption of I₂ and CH₃I, the performance of the synthesized COFs is good but not excellent. The static I₂ uptake ranks 4th and 5th compared with other 22 materials. The dynamic adsorption capacity of the synthesized COFs is also not the best. Moreover, the t_{80%} is 18 h and 24 h for the synthesized COFs, which are longer than many of other materials, representing the low adsorption dynamic performance of the two COFs. Therefore, from both the viewpoints of the features of COF structure design and the adsorption performance, this work is not recommended to be published in nature communication. Following questions are suggested to be addressed:

Response: We thank the reviewer for the critical but insightful comments, and we would like to clarify the significance of this work from the following aspects.

As indicated in our manuscript, static I₂ uptake measured in a closed system with saturated I₂ vapor at 75°C is NOT a meaningful criterion for assessing the I₂ capture performance, because these conditions are far from those in practical applications (co-existence of I₂ and organic iodides at much lower concentrations). We measured static I₂ uptakes in this study only for comparison purpose, and the ranking of our adsorbents on this metric is not important.

Adsorbing I₂ at high pressures is not challenging due to the strong intermolecular interactions of I₂. In fact, many reported adsorbents exhibit static I₂ uptakes higher than the theoretical values calculated from the pore volume and the density of iodine, because I₂ can be easily adsorbed at the external surface of the adsorbent particles and condensed in the inter-particle pores. Please refer to Fig. R1 shown in our response to Reviewer #1.

What is really challenging and largely overlooked in the literature is the simultaneous capture of I₂ and CH₃I under low concentration and dynamic conditions, which is the focus of this study. Our adsorbent design is based on these considerations. The designed adsorbent, COF-TAPT, combines high crystallinity, a large surface area, and abundant nucleophilic groups, and exhibits **a record-high**

static CH₃I adsorption capacity (1.53 g·g⁻¹ at 25 °C). In the dynamic mixed-gas adsorption with 150 ppm of I₂ and 50 ppm of CH₃I (close to the practical application condition), COF-TAPT outperformed all tested adsorbents except an ionic COF in terms of total iodine capture. These results indicate that the iodine capture performance of COF-TAPT is not “good” but “excellent”.

More importantly, the two COFs, namely COF-TAPB and COF-TAPT, exhibit the same crystal structure and textural properties, being different only in N content. Thus, they provide an unprecedented platform for systematic mechanistic studies on the different roles of textural property and N content in the adsorption of I₂ and CH₃I. Through these studies, we gained a series of new insights.

- I₂ can be relatively easily adsorbed on a variety of electron-donor sites, including various N species and aromatic moieties, by forming charge-transfer complexes and polyiodides. Therefore, the characteristics of binding sites and textural properties (e.g., surface area and pore volume) of the adsorbent both affect I₂ uptake.
- The adsorption of CH₃I occurs specifically on nucleophilic N sites with a one-to-one correspondence through N-methylation reactions to form salts and is unrelated to the textural properties of the adsorbent.
- Ionic groups can strongly promote the adsorption of I₂ but have little promotional effect on the adsorption of CH₃I.
- The CH₃I binding energy at different N sites follows the order imine N > triazine N > sp³ N.

These new insights are crucial for the understanding of I₂/CH₃I adsorption mechanisms, providing valuable guidance for the development of novel adsorbents for radioactive iodine capture and related applications.

We hope that the clarifications we offer here disentangle the true value of our work from the reviewer’s concerns.

1. Figure 5d, the I₂ transformed into I₃⁻ and I₅⁻, what’s the origin of the negative charge. If the COF framework takes one positive charge, will the COF be stable under this condition?

Response: We thank the reviewer for raising this question. It has been well documented that I₂ could be relatively easily adsorbed on a variety of electron-donor sites, including various N species and

aromatic moieties, by forming charge-transfer complexes and polyiodides.¹⁵⁻²⁴ That is, the negative charge of polyiodides originates from the functional groups of the adsorbent.

The framework charge does not affect the framework stability. Aluminosilicate zeolites have negatively charged framework interacting with extra-framework cations, while they are highly stable. Likewise, COF-TAPT exhibits high stability after I₂ adsorption, as evidenced by its ability to be reused without detectable changes in structure, porosity and I₂ adsorption capacity (see Fig. S9).

Fig. S9 Structural characterizations and regeneration ability of COF-TAPT during I₂ adsorption. **(a)** FT-IR spectra of COF-TAPT, I₂@COF-TAPT, and COF-TAPT-Re; **(b)** PXRD patterns of COF-TAPT, I₂@COF-TAPT, pure I₂, and COF-TAPT-Re; **(c)** N₂ sorption of COF-TAPT and COF-TAPT-Re; **(d)** Dynamic I₂ adsorption capacity of COF-TAPT in four successive adsorption/extraction cycles at 25 °C with 400 ppm I₂. “I₂@COF-TAPT” refers to I₂-saturated COF-TAPT. “COF-TAPT-Re” refers to regenerated COF-TAPT after one adsorption/extraction cycle.

2. The density of solid I₂ is 4.93 g/cm³, the pore volumes of COF-TAPT and COF-TAPB are lower than 1 cm³/g. Why the static uptake is high to 7~8 g/g? Whether the I₂ vapor is condensed to the outside surface of COFs when they are weighed at room temperature?

Response: We thank the reviewer for raising this insightful question.

The reviewer is correct that the measured static I₂ uptakes are higher than the theoretical values calculated from the pore volume and the density of iodine. This is a very common phenomenon (see Fig. R1 for summary data) and also why we emphasize that the static I₂ uptake value is not a meaningful criterion for performance evaluation. A plausible explanation for this phenomenon is that under such measurement conditions, there is a large amount of I₂ adsorbed at the external surface of the adsorbent particles and condensed in the inter-particle pores. We will discuss this issue shortly in a separation publication.

In the current study, we measured static I₂ uptake following the widely used protocol for comparison purpose only, whereas our focus was the ability of COF-TAPT to simultaneously capture low-concentration I₂ and CH₃I under dynamic adsorption conditions.

3. The I₂ and CH₃I are radioactive, stability of the COFs should be tested under radiation condition.

Response: We thank the reviewer for raising this point. Following the reviewer's comments, we evaluated the stability of COF-TAPT under radiation conditions. Specifically, COF-TAPT was treated with high doses (100 kGy and 200 kGy) of β-irradiation using an electron accelerator (1.0 MeV, Wasik Associates Inc., USA). The PXRD results reveal that COF-TAPT well maintained its crystallinity after irradiation, confirming its excellent irradiation stability. These results are presented in the revised manuscript as Fig. S3.

Fig. S3 PXR D patterns of COF-TAPT before and after various treatments, including acid treatments with concentrated HNO₃ aqueous solution (3M, 5M) for 48 h, and irradiation treatments with β -irradiation (100 kGy, 200 kGy) provided by an electron accelerator (1.0 MeV, Wasik Associates Inc., USA).

4. The conclusion of “The CH₃I adsorption capacity is positively correlated with the content of strong binding sites but is unrelated to the textural properties of the adsorbent.” is one-sided as the number of COFs tested in this work is small.

Response: We thank the reviewer for raising this point. Reviewer 1 also expressed the same concern, recommending “quantitative and statistical analyses”.

In order to reinforce this conclusion, we include two additional COFs materials (i.e., COF-OH-0¹⁴ and TPB-DMTP-COF¹⁵) in the analysis. The results for seven adsorbents in total yield a good linear relationship between the CH₃I uptake and the N content of the adsorbent with a reasonable coefficient of determination ($R^2=0.904$). Moreover, the irrelevance between CH₃I uptake and the BET surface area of the adsorbent is also further confirmed.

In the revised manuscript, these additional data and statistical analysis results are provided in Fig. S5, according to the reviewers’ comments.

5. The content is similar to the previously published paper *Angew. Chem. Int. Ed.* 2021, 60, 2 – 11.

Response: We thank the review for raising this concern. However, we cannot find the literature according to the information provided by the reviewer.

We suspect that the reviewer might be referring to a paper published in *Angew. Chem. Int. Ed.* (2021, 60, 19797–19803), which reported two COFs materials (TtaTfa and TpaTfa) with the same structures as COF-TAPT and COF-TAPB.

If this is the case, please kindly note that we explicitly state in the manuscript that COFs with the same structures as COF-TAPT and COF-TAPB were reported before for CO₂ adsorption (*Chemical Communications* 2017, **53**, 4242–4245; *Microporous and Mesoporous Materials* 2020, **297**, 11001). In the paper published in *Angew. Chem.*, the COFs were used for photocatalytic hydrogen evolution. Instead of claiming that the two COF materials are novel, we emphasize that we developed a new scope of application for them. The paper published in *Angew. Chem.* has been cited in the revised manuscript.

Reviewer #3:

Comments: Simultaneous capture of iodine and methyl iodide by porous materials through breakthrough test remains a challenge. As stated in the text, there is only one COF that covers this feature. In this work, Han et al prepared two comparable COFs but with the difference in nitrogen atom, thus, giving an insight into the relationship between I₂, CH₃I uptake and various nitrogen atoms. Effective and simultaneous capture of iodine and methyl iodide were observed for the nitrogen-rich COFs. The results are very interesting. The preparation of this work was careful. But some points should be considered before publication.

Response: we are truly grateful to the reviewer for his/her appreciation of our work.

1. "To the best of our knowledge, there is only one COF material (SCU-COF-2) evaluated for both I₂ and CH₃I adsorption". In fact, Han and co-workers has also reported a MOF for such use. Thus, please revise this sentence.

Response: We thank the reviewer for raising this point. The reviewer is correct that modified MOF MIL-101 has been tested for simultaneous capture of I₂ and CH₃I (Nature Communications, 2017, 8, 485). However, in our manuscript, the statement above was placed in the context of COF materials. It is difficult for us to include a MOF-based adsorbent in this statement without affecting the narrative flow. Therefore, we prefer not to modify this statement.

2. Although the author has made deep research in both I₂ and CH₃I breakthrough test, however, as investigated in their previous work, simultaneous capture of iodine and methyl iodide at high temperature such as 423 K is suggested to be investigated, especially in the presence of HNO₃.

Response: We thank the reviewer for this insightful suggestion. We have measured the I₂ and CH₃I uptakes of COF-TAPT and COF-TAPB under dynamic condition at 75 °C. It turned out that for both materials, increasing the measurement temperature from 25 °C to 75 °C leads to significant decrease in I₂/CH₃I adsorption capacity (**Fig. R3**). These results indicate that COF-TAPT and COF-TAPB are unsuitable for iodine capture at elevated temperatures. Therefore, this study has been focused on the I₂ and CH₃I adsorption behaviors at room temperature.

Fig. R3 Dynamic I₂ adsorption and CH₃I adsorption performances. (a) I₂ breakthrough profiles of COF-TAPB and COF-TAPT at 25 °C and 75 °C. The concentration of I₂ in the carrier gas is 400 ppm. (b) Dynamic I₂ uptake values of COF-TAPB and COF-TAPT at 25 °C and 75 °C. (c) CH₃I breakthrough profiles of COF-TAPB and COF-TAPT at 25 °C and 75 °C. The concentration of CH₃I in the carrier gas is 200,000 ppm. (d) Dynamic CH₃I uptake values of COF-TAPB and COF-TAPT at 25 °C and 75 °C.

3. How about the irradiation stability of this material?

Response: Reviewer 2 raised the same question. Following the reviewers' comments, we evaluated the stability of COF-TAPT under radiation conditions. Specifically, COF-TAPT was treated with high doses (100 kGy and 200 kGy) of β -irradiation using an electron accelerator (1.0 MeV, Wasik Associates Inc., USA). The PXRD results indicate that COF-TAPT well maintained its crystallinity after irradiation, confirming its excellent irradiation stability. These results are presented in the revised manuscript as Fig. S3.

4. How about the stability of this material under HNO₃?

Response: We thank the reviewer for raising this question. The PXRD results indicate that treatment with concentrated HNO₃ aqueous solution (3M or 5M) for 48 h did not significantly reduce the crystallinity of COF-TAPT. In the revised manuscript, these results are shown in Fig. S3.

5. The pore size is not given in Fig. 1.

Response: As suggested by the reviewer, we have updated Fig. 1 to include the pore size distribution profiles for COF-TAPB and COF-TAPT, which are derived from the N₂ sorption isotherms using the NLDFT method.

6. In Fig. 4, the uptake is based on weighing method, thus, error bar is suggested.

Response: Following this suggestion, Fig. 4 has been updated with error bars, which were determined based on the results of three parallel measurements.

Fig. 4 Comparison of different adsorbents in dynamic adsorption capacity measured from (a) single-component I₂ at 150 ppm, (b) single-component CH₃I at 50 ppm, (c) mixed-component (150 ppm I₂ and 50 ppm CH₃I) breakthrough experiments performed at 25 °C. The error bars are the standard deviations from three parallel measurements.

7. The adsorption of anion ion or anionic dye is suggested to further confirm the CH₃I adsorption through methylation reaction.

Response: Following the reviewer's suggestion, we tested the anion exchange capacity of CH₃I-saturated COF-TAPT (CH₃I@COF-TAPT) using Cr₂O₇²⁻ as a model anion. Specifically, 50 mg of CH₃I@COF-TAPT was added to a 200 mL beaker containing 100 mL of potassium dichromate (K₂Cr₂O₇, 0.1 mM) solution. The mixture was kept stirring at 25 °C; at specific time points, 2 mL of sample was

taken from the mixture, separated with a 0.22 mm nylon membrane filter, and measured by UV-vis spectroscopy. The results indicate that $\text{CH}_3\text{I}@\text{COF-TAPT}$ can efficiently capture $\sim 90\%$ Cr(VI) within 30 min and $\sim 95\%$ Cr(VI) after 180 min, which is consistent with the apparent discoloration of the solution during the anion-exchange process. These results are shown in Fig. S13 in the revised manuscript to provide additional evidence that CH_3I is adsorbed through methylation reactions.

Fig. S12 UV-vis absorption spectra of 0.1 M $\text{K}_2\text{Cr}_2\text{O}_7$ aqueous solution collected at specific times during the anion exchange process. The CH_3I saturated COF-TAPT ($\text{CH}_3\text{I}@\text{COF-TAPT}$) was used as anion exchanger. The insets are photo pictures, showing the apparent discoloration of the solution during the process.

8. How about the recycle use of simultaneous capture of iodine and methyl iodide at ppm level.

Response: Following this suggestion, we investigated the reusability of COF-TAPT for simultaneous capture of iodine and methyl iodide at ppm level (150 ppm I_2 and 50 ppm CH_3I at 25 °C). The results revealed that the adsorption capacity of COF-TAPT for total iodine (I_2 and CH_3I) hardly changed in four successive adsorption/extraction cycles. These results are presented in Fig. S11 in the revised manuscript.

Fig. S11 Dynamic adsorption capacity of COF-TAPT with a stream of 150 ppm I₂ and 50 ppm CH₃I in four successive adsorption/extraction cycles at 25 °C.

Reference

1. Svensson, P. H., Kloo, L. Synthesis, structure, and bonding in polyiodide and metal iodide–iodine systems. *Chemical Reviews* **103**, 1649–1684 (2003).
2. Nandanwar, S. U., Coldsnow, K., Utgikar, V., Sabharwall, P., Aston, D. E. Capture of harmful radioactive contaminants from off-gas stream using porous solid sorbents for clean environment—A review. *Chemical Engineering Journal* **306**, 369–381 (2016).
3. Xie, W., Cui, D., Zhang, S.-R., Xu, Y.-H., Jiang, D.-L. Iodine capture in porous organic polymers and metal–organic frameworks materials. *Materials Horizons* **6**, 1571–1595 (2019).
4. Taghipour, F., Evans, G. J. Radiolytic organic iodide formation under nuclear reactor accident conditions. *Environmental Science & Technology* **34**, 3012–3017 (2000).
5. Chebbi, M., Azambre, B., Volkringer, C., Loiseau, T. Dynamic sorption properties of Metal-Organic Frameworks for the capture of methyl iodide. *Microporous and Mesoporous Materials* **259**, 244–254 (2018).
6. Taylor, D. M. The radiotoxicology of iodine. *Journal of Radioanalytical Chemistry* **65**, 195–208 (1981).
7. Goldsmith, J. R., et al. Juvenile hypothyroidism among two populations exposed to radioiodine. *Environmental Health Perspectives* **107**, 303–308 (1999).
8. Michel, R., et al. Iodine-129 in soils from Northern Ukraine and the retrospective dosimetry of the iodine-131 exposure after the Chernobyl accident. *Science of The Total Environment* **340**, 35–55 (2005).
9. Katritzky, A. R., Meth-Cohn, O., Rees, C. W. *Comprehensive Organic Functional Group Transformations: Synthesis: carbon with one heteroatom attached by a single bond*. Elsevier (1995).
10. Ashokkumar, M., Grieser, F. Ultrasound assisted chemical processes. *Reviews in Chemical Engineering* **15**, 41–83 (1999).
11. Siddique, M., Farooq, R., Khan, Z. M., Khan, Z., Shaukat, S. Enhanced decomposition of reactive blue 19 dye in ultrasound assisted electrochemical reactor. *Ultrasonics Sonochemistry* **18**, 190–196 (2011).

12. Dobaradaran, S., et al. Catalytic decomposition of 2-chlorophenol using an ultrasonic-assisted Fe₃O₄-TiO₂@ MWCNT system: Influence factors, pathway and mechanism study. *Journal of colloid and interface science* **512**, 172–189 (2018).
13. Sheikhmohammadi, A., Asgari, E., Nourmoradi, H., Fazli, M. M., Yeganeh, M. Ultrasound-assisted decomposition of metronidazole by synthesized TiO₂/Fe₃O₄ nanocatalyst: Influencing factors and mechanisms. *Journal of Environmental Chemical Engineering* **9**, 105844 (2021).
14. Xie, Y., et al. Ionic Functionalization of Multivariate Covalent Organic Frameworks to Achieve an Exceptionally High Iodine-Capture Capacity. *Angewandte Chemie International Edition* **60**, 22432–22440 (2021).
15. Wang, P., Xu, Q., Li, Z., Jiang, W., Jiang, Q., Jiang, D. Exceptional iodine capture in 2D covalent organic frameworks. *Advanced Materials* **30**, 1801991 (2018).
16. Niu, T.-H., Feng, C.-C., Yao, C., Yang, W.-Y., Xu, Y.-H. Bisimidazole-Based Conjugated Polymers for Excellent Iodine Capture. *ACS Applied Polymer Materials* **3**, 354–361 (2020).
17. Xu, M., Wang, T., Zhou, L., Hua, D. Fluorescent conjugated mesoporous polymers with N, N-diethylpropylamine for the efficient capture and real-time detection of volatile iodine. *Journal of Materials Chemistry A* **8**, 1966–1974 (2020).
18. He, L., et al. A nitrogen-rich covalent organic framework for simultaneous dynamic capture of iodine and methyl iodide. *Chem* **7**, 699–714 (2021).
19. Jiang, X., et al. Topochemical Synthesis of Single-Crystalline Hydrogen-Bonded Cross-Linked Organic Frameworks and Their Guest-Induced Elastic Expansion. *Journal of the American Chemical Society* **141**, 10915–10923 (2019).
20. Geng, T., Chen, G., Xia, H., Zhang, W., Zhu, Z., Cheng, B. Poly {tris [4-(2-thienyl) phenyl] amine} and poly [tris (4-carbazoyl-9-yl phenyl) amine] conjugated microporous polymers as absorbents for highly efficient iodine adsorption. *Journal of Solid State Chemistry* **265**, 85–91 (2018).
21. Guo, X., et al. Colyliform Crystalline 2D Covalent Organic Frameworks with Quasi-3D Topologies for Rapid I₂ Adsorption. *Angewandte Chemie International Edition* **132**, 22886–22894 (2020).

22. Geng, T., Zhang, W., Zhu, Z., Kai, X. Triazine-based conjugated microporous polymers constructing triphenylamine and its derivatives with nitrogen as core for iodine adsorption and fluorescence sensing I₂. *Microporous and Mesoporous Materials* **273**, 163–170 (2019).
23. Xiong, S., Tang, X., Pan, C., Li, L., Tang, J., Yu, G. Carbazole-bearing porous organic polymers with a mulberry-like morphology for efficient iodine capture. *ACS Applied Materials & Interfaces* **11**, 27335–27342 (2019).
24. Geng, T., Zhang, C., Liu, M., Hu, C., Chen, G. Preparation of biimidazole-based porous organic polymers for ultrahigh iodine capture and formation of liquid complexes with iodide/polyiodide ions. *Journal of Materials Chemistry A* **8**, 2820–2826 (2020).

Reviewers' Comments:

Reviewer #1:

Remarks to the Author:

Authors have fully addressed my previous concerns during their revision process. In addition, I have gone through comments from reviewers 2 and 3 and authors' corresponding response and revision. With those, I can come with a solid conclusion that authors have made adequate improvement for their manuscript based on these comments. I especially agree with authors' arguments of "As indicated in our manuscript, static I2 uptake measured in a closed system with saturated I2 vapor at 75°C is NOT a meaningful criterion for assessing the I2 capture performance, because these conditions are far from those in practical applications" and "What is really challenging and largely overlooked in the literature is the simultaneous capture of I2 and CH3I under low concentration and dynamic conditions, which is the focus of this study". I hope reviewer 2 would take this message and if they are also investigators in this field, they should also look into property under practical scenario instead of irrelevant experimental condition. This work should be published in its current form.

Reviewer #2:

Remarks to the Author:

The author answered each question seriously. Although the author emphasizes that the feature of this work is the simultaneous adsorption of I2 and CH3I, the so called excellent performance is not an overwhelming superiority than other materials. The two COFs are also previously reported. The mechanism in the adsorption is also not novel. The little higher adsorption amount of I2 and CH3I is insufficient to support the publication in Nature Commun.

REVIEWERS' COMMENTS

Reviewer #1 (Remarks to the Author):

Authors have fully addressed my previous concerns during their revision process. In addition, I have gone through comments from reviewers 2 and 3 and authors' corresponding response and revision. With those, I can come with a solid conclusion that authors have made adequate improvement for their manuscript based on these comments. I especially agree with authors' arguments of "As indicated in our manuscript, static I₂ uptake measured in a closed system with saturated I₂ vapor at 75°C is NOT a meaningful criterion for assessing the I₂ capture performance, because these conditions are far from those in practical applications" and "What is really challenging and largely overlooked in the literature is the simultaneous capture of I₂ and CH₃I under low concentration and dynamic conditions, which is the focus of this study". I hope reviewer 2 would take this message and if they are also investigators in this field, they should also look into property under practical scenario instead of irrelevant experimental condition. This work should be published in its current form.

Response: We are grateful to the reviewer for indicating the true value of our work.

Reviewer #2 (Remarks to the Author):

The author answered each question seriously. Although the author emphasizes that the feature of this work is the simultaneous adsorption of I₂ and CH₃I, the so called excellent performance is not an overwhelming superiority than other materials. The two COFs are also previously reported. The mechanism in the adsorption is also not novel. The little higher adsorption amount of I₂ and CH₃I is insufficient to support the publication in Nature Commun.

Response: We thank the reviewer for taking the time to evaluate our work. We regret that this reviewer is not fully convinced by the additional data and clarifications we provided.